# Evaluation of a Non-Face-to-Face Multidisciplinary Health Care Model in a Population with Rheumatoid Arthritis Vulnerable to COVID-19 in a Health Emergency Situation [note 1]

**DOI:** 10.3390/healthcare9121744

**Published:** 2021-12-17

**Authors:** Pedro Santos-Moreno, Gabriel-Santiago Rodríguez-Vargas, Rosangela Casanova, Jaime-Andrés Rubio-Rubio, Josefina Chávez-Chávez, Diana Patricia Rivera-Triana, Ruth Alexandra Castiblanco-Montañez, Sandra Milena Hernández-Zambrano, Laura Villareal, Adriana Rojas-Villarraga

**Affiliations:** 1BIOMAB IPS, Bogota 110231, Colombia; laura.villarreal@biomab.co; 2Research Division, Fundación Universitaria de Ciencias de la Salud-FUCS, Bogota 111221, Colombia; gsrodriguez@fucsalud.edu.co (G.-S.R.-V.); rcasanova@fucsalud.edu.co (R.C.); jandresr2@gmail.com (J.-A.R.-R.); jchavez@fucsalud.edu.co (J.C.-C.); dprivera@fucsalud.edu.co (D.P.R.-T.); sarojas@fucsalud.edu.co (A.R.-V.); 3Nurse Faculty, Fundación Universitaria de Ciencias de la Salud-FUCS, Bogota 111221, Colombia; racastiblanco@fucsalud.edu.co (R.A.C.-M.); smhernandez3@fucsalud.edu.co (S.M.H.-Z.)

**Keywords:** COVID-19, Latin America, rheumatoid arthritis, telemedicine

## Abstract

This study evaluated a non-face-to-face-multidisciplinary consultation model in a population with rheumatoid arthritis (RA) during the COVID-19 pandemic. This is an analytical observational study of a prospective cohort with simple random sampling. RA patients were followed for 12 weeks (Jul–Oct 2020). Two groups were included: patients in telemedicine care (TM), and patients in the usual face-to-face care (UC). Patients could voluntarily change the care model (transition model (TR)). Activity of disease, quality of life, disability, therapeutic adherence, and self-care ability were analyzed. Bivariate analysis was performed. A qualitative descriptive exploratory study was conducted. At the beginning, 218 adults were included: (109/TM-109/UC). The groups didn’t differ in general characteristics. At the end of the study, there were no differences in TM: (*n* = 71). A significant (*p* < 0.05) decrease in adherence, and increase in self-care ability were found in UC (*n* = 18) and TR (*n* = 129). Seven patients developed COVID-19. Four categories emerged from the experience of the subjects in the qualitative assessment (factors present in communication, information and communication technologies management, family support and interaction, and adherence to treatment). The telemedicine model keeps RA patients stable without major differences compared to the usual care or mixed model.

## 1. Introduction

Since December 2019, the world has faced a public health emergency caused by severe acute respiratory syndrome coronavirus 2 (SARS-CoV2) [1], which caused a reconsideration of the entire healthcare system, and the invention of new ways to help patients while minimizing infection risk. Coronavirus 19 (COVID-19) generated a major challenge for third world countries like those in Latin America (LA), especially because [2] of deficient health care access, substantial poverty, and a need to implement new technologies amid these difficulties. Telemedicine became important in different areas of medicine, including rheumatology. Different platforms and technology tools began to be more frequently used to see rheumatologic patients at a distance, and provide safe, cost-effective, and convenient care for both the patient and the treating medical team [3,4]. The impact of COVID-19 on rheumatology practice was demonstrated in some countries by a decrease of up to 65% in outpatient visits [5].

Recently, studies were carried out to evaluate the impact of telemedicine on RA patients during the pandemic, since the therapies used to treat this pathology generally predispose RA patients, who are at high risk to infections [6,7]. The main concern has been to minimize exposure to SARS-CoV2, and continue the strict medical follow-up these patients need [8].

Prior to the pandemic, the importance of using telemedicine to teach RA patients strategies for adhering to their medication had been reported [9]. Assessing the level of adherence under telemedicine during the pandemic will help determine subsequent measures to follow and ensure good clinical outcomes. Some preliminary studies suggest that most patients continued their therapy in spite of the pandemic [10].

There are prior studies evaluating the outcomes for RA patients in terms of disease activity or quality of life under telemedicine. For instance, a study conducted in Denmark in 2015 proved that for RA patients with low disease activity or in remission, telemedicine follow-up was not inferior to conventional outpatient follow-up, and was cost-effective [11]. Other studies have shown that RA patients were satisfied with the telemedicine attention, it cost the health system less, and worked in favor of the patients [12,13]. Moreover, patient-reported outcomes for telemedicine care have been shown to be similar to usual care, with significant cost and distance savings [14]. However, it cannot be assumed that telemedicine completely replaces face-to-face attention. Instead, some researchers think it should be used to complement and bolster in emergencies like the current one [15].

Generally, most of the telemedicine RA studies have been done in European countries, China, Africa, India, and the United States [16], with limited information on LA or Colombian RA patients [2,17,18]. Little evidence of the telemedicine model for RA patients in Latin America under the COVID-19 circumstances has been documented.

The main objective of this study was to evaluate a non-face-to-face multidisciplinary consultation model in a population with RA, highly vulnerable to COVID-19, during a health emergency situation, and the effectiveness of their results. In addition, a qualitative analysis was performed on RA patients’ and professionals’ experiences in implementing that model, and to identify what motivated the group of RA patients who did not participate in it.

This is the first study done of LA and Colombian RA patients. Whereas the telemedicine model is evaluated using telephone calls as the only tool, studies carried out in other countries used different platforms and tools, such as video calling.

## 2. Materials and Methods

### 2.1. Study Design

This was a prospective, analytical, observational, cohort study that used a mixed analysis (quantitative and qualitative) to compare a telemedicine model (TM) to the usual face-to-face care model (UC) for follow-up on RA patients in a health emergency.

### 2.2. Setting

Patients were followed for 12 weeks (July to October of 2020) at a specialized health care center (BIOMAB IPS) in Bogota, Colombia (Clinical trial registration NCT04768413).

### 2.3. Participants

The eligibility criteria was patients over 18 years of age with a confirmed RA diagnosis (International Classification of Diseases: M069, M059, M060). They were classified as having RA if they fulfilled the American College of Rheumatology classification criteria for rheumatoid arthritis [19] They also had access to a telephone, or information and communication technologies (ICTs). The exclusion criteria was patients who, due to cognitive conditions, were unable to provide the reliable information needed to take part in the study.

Two groups of patients were included. Group A TM model: a group of patients who voluntarily chose the clinic’s TM consultation, and received remote, multidisciplinary team care (by phone using the Walter Bridge^®^ platform) at home. Group B UC Model: patients who continued the usual face-to-face doctor visit since isolation measures allowed them to make these trips, and they received care from a multidisciplinary team regularly. Patients could voluntarily change the care model whenever they desired. A database was constructed from the individuals who met the inclusion criteria, and did not meet the exclusion criteria from the general database of the institution. They were invited by phone, and those who accepted were included in the next face-to-face or telephone consultation. This corresponded to visit 1, in which they signed the informed consent. This was done consecutively until the estimated number derived from the sample size calculation was completed. For the qualitative component: adult RA patients in the TM model were evaluated on at least two occasions, patients treated in UC model on at least two occasions, and health care professionals (HCP) who have carried out at least 25 telemedical consultations in the health emergency context.

### 2.4. Variables

#### 2.4.1. Sociodemographic Data and Clinical Characteristics

Electronic medical records were reviewed (baseline) for information on demographic characteristics (gender, age, marital status, education, and current occupational status) and clinical history (comorbidities, previous surgical procedures or infections, erosivity, polyautoimmunity, extra articular manifestations, and age at RA diagnosis). Furthermore, information on currently prescribed medications and modifications during follow-up was obtained.

#### 2.4.2. Outcome Measurements and Follow-Up Strategy

Disease activity: patient activity scale (PAS) [20], participants’ pain using visual analogue scale 0–10 cm (VAS), patient global assessment (PtGA) 0–10 cm, disease activity by doctor using VAS 0–10 cm scores in both groups, and disease activity score with 28-joint counts (DAS28) [21] in the UC group were evaluated at baseline, week 6, and week 12. Disability was evaluated (baseline, week 6, and week 12) using the health assessment questionnaire (HAQ) [22] for both groups. The patient activity scale was calculated for both groups by multiplying the HAQ by 3.33, and dividing the sum of VAS pain and HAQ by 3 (categories considered for disease activity: remission ≤0.25, low ≤3.7, moderate <8.0, and high ≥8.0). For patients using the UC model, the disease activity level was interpreted as remission (DAS28 < 2.6), low (2.6 ≤ DAS28 < 3.2), moderate (3.2 ≤ DAS28 ≤ 5.1), or high (DAS28 > 5.1). For HAQ calculations, average scores from 0–1 represented “mild to moderate difficulty”, 1–2 meant “moderate to severe disability”, and 3–2 indicated “severe to very severe disability”. 

Quality of life was assessed with the European Quality of Life-5 dimension (EQ-5D-3L) instrument [23,24]. Medication adherence was evaluated with the four item Morisky-Green Levine Medication Adherence Scale (MGLS) [25], and both groups were evaluated for self-care agency with the Appraisal of Self-Care Agency Scale-Revised (ASAS-R) [26] at baseline, and week 12 by phone. For EQ5 calculations, an overall index score (that measures from the lowest (worst) to the highest (best) score) was calculated for each patient and time point. We used the time trade-off (TTO) valuation technique based on the list of currently available value sets for the EQ-5D-3L. Most of the EQ-5D value sets have been obtained from a representative sample of the general population, thereby ensuring that they represent the societal perspective. Therefore, the value sets from Spain were chosen since none are currently available for Colombia. For the calculation, the index value set calculator was implemented by means of the STATA^®^ value set-Spain-EQ-5D-3L command available [27]. State of health was also measured on the vertical VAS (score 0 to 100), where higher scores equal better health status.

Three levels of medication adherence based on the MGLS score were obtained: high, medium, and low adherence, with 0, 1–2, and 3–4 points, respectively. Two levels of medication adherence based on the MGLS score were obtained: high and low adherence, with 0 and 1–4 points, respectively.

Regarding ASAS-R, which has a total of 15 questions in 3 areas (the area lacking self-care agency was reversely coded), and based on a 5-point Likert scale, a higher total score was interpreted with a higher level of self-care agency.

Information regarding a laboratory or clinically confirmed diagnosis of SARS-CoV-2 was extracted from clinical records at week 6 and 12; as were outcomes related to COVID-19 (hospitalization or death).

### 2.5. Qualitative Analysis and Collection

A qualitative, descriptive, exploratory study was done. Semi-structured interviews (telephone or video-based depending on participant preferences) were carried out to analyze the experiences of RA patients, and their HCP with respect to the implementation of a non-face-to-face multidisciplinary consultation model in the health emergency due to COVID-19. A semi-structured thematic script with exploratory questions (Appendix A) based on the study dimensions for each of the three specific groups was used: group 1 (face-to-face consultation patients), group 2 (teleconsultation patients), group 3 (health professionals who work at BIOMAB IPS). The script was used to guarantee coherence with the study dimensions to be explored, and to facilitate progressive analysis. However, the interviewer facilitated the discussion and emergence of new topics to understand the issue under study.

All interviews were conducted by researchers trained in qualitative designs, and audio-recorded with the participants’ prior informed consent. The qualitative results are supported by a content analysis consistent with the qualitative design. The Taylor–Bogdan [28] proposal, as adapted by Amezcua and Gálvez [29], was followed, in addition to international criteria for the analysis and rigor of qualitative studies [30]. This consisted of data preparation, topic discovery, data coding, and relativization. The data from each interview were analyzed in pairs to triangulate the information. After that, a global triangulation of the results of the interviews was carried out with a third researcher in which criteria of confirmability, credibility, and consistency of the data were assessed to guarantee the rigor of the study.

### 2.6. Data Sources

Three expert researchers collected all the information (during clinical visits using TM or UC model, baseline, week 6 and 12) using the Research Electronic Data Capture (REDCap) platform [(https://www.project-redcap.org/ (accessed on 1 July 2020)]. EQ-5D-3L, MGLS, ASAS-R scales were applied (baseline and week 12) to both groups by phone (the same day of the consultation or within 1 week).

### 2.7. Statistics Analysis

#### 2.7.1. Sample Size Calculation

A sample size for mean differences was calculated using the data reported by Chew Li-Ching et al. [31], a minimum effect size of 0.40 [32], and a two-tailed analysis of covariance with a minimum power of 80% and a 1:1 ratio. The calculation yielded 77 patients per group. However, the sample size was increased to 110 for each group in anticipation of possible loss or patients’ voluntary change of group during follow-up. However, the final sample included 109 patients, because one in each group did not meet the eligibility criteria. For the calculation, the statistical program OpenEpi v3.01 was used.

#### 2.7.2. Statistical Plan

Data was exported from REDcap in Excel format, and statistical software Stata version 13^®^ was used to analyze and obtain the results. Descriptive analyses were done using absolute and relative frequencies for qualitative variables, and central tendency measures (mean or median) and dispersion (standard deviation (SD) or interquartile range (IQR)) for quantitative variables based on the distribution of normality test results. For an analysis of the outcomes (change from baseline), the chi-squared test in expected values greater than 5, and Fisher’s exact test in values less than 5 were used for the qualitative variables. For the quantitative variables, either the Wilcoxon test or T test or Mann–Whitney U test was used depending on the normality of their distribution (change from baseline). Statistical differences were determined to be significant at *p*-values less than 0.05. For COVID-19-related variables, proportions were calculated for outcomes in each group.

The calculation of the sample size had a selection bias since there were no previous studies with evidence from patients with RA and COVID-19 to estimate the ratio between exposed and unexposed. Therefore, a possible approximate 1:1 ratio between exposed and unexposed was used in the calculation.

The main measurement bias in this study came from rapid changes in knowledge regarding COVID-19, as this necessarily affected the measurement of the effect on the groups in relation to the evolution of the disease in real time. This bias was addressed by presenting the results based on the current knowledge as of the time of publication.

As an observational study of a disease with a low prevalence, another bias is related to the follow-up of the groups. Although the sample size is adequate, the short duration of follow-up did not allow losses to be properly assessed and compensated for. Therefore, different analyses were conducted based on the change from one model to another.

This study, including procurement of informed consent electronically and by phone, was approved by the Ethics Committee for Research on Human Beings (CEISH) by Hospital de San José HSJ-FUCS (Identifier: # 02-062020).

## 3. Results

### 3.1. Sociodemographic and Baseline Characteristics

There were 218 patients with a confirmed diagnosis of RA. This included 109 in the TC model, and 109 in UC. The groups did not show differences in terms of sociodemographic characteristics, except in occupation (*p* < 0.001) (Table 1).

Regarding comorbidities, osteoarthrosis predominated in both groups, followed by osteoporosis, with statistically significant differences in concomitant fibromyalgia, which was higher in the UC group (*p* = 0.01), although frequency was low in both. Patients in the TM group (84%) had more surgical antecedents (*p* < 0.001).

There were no differences between the classes of medications the patients were taking at the baseline when comparing TM and UC models.

### 3.2. Disease Baseline Characteristics

Age at symptom onset and at diagnosis were similar in both groups. Polyautoimmunity was present in 16.5% of the patients with TM doctor visits, and in 10.1% of those who went in-person (no statistically significant differences), with a higher frequency of Sjogren’s syndrome in both. Pulmonary compromise was the most frequent extra-articular manifestation, although frequency was low in both. Erosivity was present in 49% of the patients seen by TM, and 40.4% in UC (Table 1).

### 3.3. Clinical Outcomes

Figure 1 shows the flow chart of patients in each modality during follow-up. At the end, there were three groups depending on the patients’ transition between models. Results are shown for the three groups that had enough individuals to do an analysis (no transition—UC (*n* = 18) or TM (*n* = 71); transition between models [(TR) *n* = 129]). Comparisons between visits for the entire group are shown in Table 2. The only variable that changed significantly when the baseline was compared to the end of follow-up was the capacity for agency and self-care. This showed a significant increase on ASAS-R (*p* = 0.0001) (Table 2). The specific outcomes are described in the following paragraphs.

#### 3.3.1. Telemedicine Model Outcomes

Note that out of 109 patients under TM, 71 still remained in this modality by week 12. Disease activity data for nine patients were lost to follow-up at week 12.

The median PGA remained constant for the three visits, whereas the median VAS pain tended to decrease (*p* = 0.125). Similarly, the PAS tended to decrease without statistical significance (*p* = 0.2252) (Table 2). Furthermore, HAQ measurements showed most patients presented mild to moderate difficulty throughout follow-up.

There were no changes in EQ-5D-3L (overall index or VAS) and ASAS-R calculations, whereas adherence (MGLS) tended to increase without statistical significance (*p* = 0.121). The mean TTO utility score, estimated as 0.8 [95% CI: 0.6–0.9] in visit 1 and visit 3, for the TM group was the same for both visits (Table 3).

#### 3.3.2. Usual Care Model Outcomes

Of the 109 patients at baseline, 18 remained exclusively in UC throughout the follow-up (0–6 and 12 weeks), and the rest transited between the two models (Figure 1).

There was a non-significant trend to decrease in the PAS score and VAS pain. The PGA and HAQ variables were stable for the three visits. The DAS-28 presented a slight non-significant declining trend beginning with moderate activity (66% in remission or moderate), then in remission (66%), and ending with low (26.7%) disease activity (Table 3).

The EQ5 VAS component presented a slight non-significant deterioration, as well as a non-significant improvement level in the EQ5 general index score. The mean TTO utility score was estimated at 0.7 (95% CI: 0.6–0.9) during visit 1, and 0.8 (95% CI: 0.6–0.9) during visit 3 without significant change (*p* = 0.77). Self-care improved significantly, as demonstrated by a higher score on the ASAS-R scale (*p* = 0.0077). Conversely, adherence measured by the MGLS scale decreased significantly (*p* = 0.006).

#### 3.3.3. Transition between Model (TR) Group Outcomes

There were 129 patients that transited between the two models (UC and TM), resulting in eight sub-groups based on each beginning model (Figure 1).

The entire group showed a significant increase (Table 3) in self-care as measured by the ASAS-R scales in the entire group (*p* = 0.001). Conversely, adherence in this group of patients measured through the MGLS scale decreased significantly (*p* = 0.03). There were no nonsignificant changes in the other clinical variables during follow-up.

#### 3.3.4. Pharmacological Outcomes

Regarding medication, comparing baseline with follow-up visits, there was a significant decrease in the number of patients receiving methotrexate (*p* = 0.027) when the whole group was analyzed (Table 2). Other medications showed no significant changes. There was a significant decrease in the number of patients who received methotrexate during follow-up in the UC group at baseline, and who were evaluated by TM in visits 2 and 3 (*p* = 0.044). No other differences were found when specific medications were analyzed (Appendix A)**.**

Analgesic prescriptions increased significantly (*p* = 0.001) in the TM group (Appendix A). Likewise, when changes made in the second and third visit of the whole group were analyzed (Appendix A), there was a significant increase in dose or analgesic prescription (*p* = 0.023).

#### 3.3.5. COVID-19-Related Outcomes

Seven patients, four women and three men, developed COVID-19 (3.2%). Five patients were transiting between the models, and only two patients were under TM and UC models, respectively. Median age was 66 years (Range 59–81). Three patients were receiving biological/targeted synthetic disease-modifying antirheumatic drugs-b/tsDMARDs (etanercept, rituximab, and golimumab, respectively). All patients were receiving oral glucocorticoids. Six were receiving conventional synthetic-csDMARDs. All seven had several comorbidities, but the two most prevalent diseases were arterial hypertension and osteoarthritis. The complete RA medication plan is shown in Appendix A.

All the patients were confirmed by PCR COVID test. One patient, using TM exclusively (female/66 years), was hospitalized, but had a complete recovery. One man (73 years) was hospitalized for 7 days, and died of pulmonary COVID-19 complications (under TR model, lost to follow-up on the third visit). Complete information regarding these patients is shown in Appendix A.

### 3.4. Qualitative Results

Thirty-seven interviews were done, and twenty-nine (78.3%) corresponded to RA patients (69% were done by TM, and 31% through UC; and 8 (21.6%) corresponded to HCP (general practitioner [*n* = 3], rheumatologist [*n* = 2], and one each with internal medicine, psychiatrist, and psychologist HCP)).

With the information obtained from the participants’ responses during the interviews, categories of the patients and professionals’ experience were created with respect to a scenario of high vulnerability and uncertainty derived from the pandemic (Table 4): factors present in communication, ICT management, family support and interaction, and adherence to treatment. For patients, mental health, pain, functional dependence, and quality of life were the most affected dimensions. Resilience mechanisms, such as adaptation and self-care measures, emerged to minimize risks.

#### 3.4.1. Factors Present in Communication

The group of patients and professionals who were interviewed stated that communication was a fundamental part of health care since it guaranteed the professional’s and patient’s understanding of information, and made it possible to clarify doubts. It also enabled a relationship that facilitated identification of each patient’s needs in order to pro-mote personalized attention. Therapeutic communication is usually mediated by body and verbal language. However, teleconsultation through a telephone call limits physical and visual contact. According to health personnel, this context should further the strengthening of people’s skills in empathy, assertive communication, problem solving, carrying out an in-depth investigation into the needs of the person, and understanding subjectivity as part of the therapeutic relationship.

#### 3.4.2. ICT Management

TM care has been justified by the reduction in the risk of contagion, access to medical care for people in dispersed areas, the economic savings due to not having to travel, and time management for people who have to work. Both the patients and professionals using it have expressed acceptance. Regarding ICT management, clinical factors (disease activity, hearing impairments, and mental illnesses such as Alzheimer’s) and social determinants (digital literacy, social support network, access to mobile devices, internet service), which may define the relevance of TM due to their influence on the use of digital tools, were identified. Therefore, it is essential to assess the individual needs and the context of the patient in order to set up the most appropriate care modality. Among the clinical factors, the following stand out: hearing disabilities, and mental illnesses such as Alzheimer’s. Regarding the social factors, digital literacy, difficulties in accessing mobile devices, lack of internet service, or difficulties in telephone reception are identified.

#### 3.4.3. Family Support and Interaction

The role of the caregiver is generally assumed by a family member who offers support and accompaniment throughout the health care process. For example, they help solve problems with virtual documentation, as well as drug delivery and clinical examinations. When patients do not have this support network, the clinic’s policy is to name an advocate for such patients, particularly elderly ones, to facilitate communication with healthcare workers, and understanding of the entire care process.

#### 3.4.4. Adherence to Treatment

In RA, adherence to treatment is of vital importance to improve symptomatology and perception of well-being. Regarding adherence, the patients mentioned several advantages of telemedicine, such as: continued monitoring and follow-up of patients who, due to conditions derived from isolation measures, could not go to the IPS; reduction of no-shows at scheduled appointments; improved opportunity to get appointments with specialists; improved adherence to treatment due to the implementation of measures that allowed medication to be sent to the patient’s home.

Most patients with RA had high levels of dependency, so having telemedicine, an alternative solution, available eliminated several problems, including: travel time, costs, a personal companion, etc. This was described as beneficial for the patients since it improved their time management and health care.

#### 3.4.5. Patients affected Dimensions

For patients, mental health, pain, functional dependence, and quality of life were the most affected dimensions. Resilience mechanisms, such as adaptation and self-care measures to minimize risks, emerged. These findings are described in the fifth category, defined as experiences of patients and healthcare personnel in the COVID-19 period. Some of the people who were interviewed expressed fear of contracting the virus, and just the fact of going to a hospital and not seeing their family again generated anxiety. However, another group of people said that the best way to deal with this contingency was to remain calm, and adapt to changes. Feelings are exacerbated when people think about the possible contagion by SARS-CoV-2 because they or their families have to leave their places of residence, and use public transportation which increases the possibility of contagion. According to the reports regarding people interviewed, these circumstances enhance their self-care practices.

Patient satisfaction with the good service provided during the consultation con-tributed to the professional–patient relationship. In this respect, expressing gratitude enhanced individual well-being, since the feeling of gratitude was an individual experience that was manifested as satisfaction with the good care received from the health personnel, and thus, the patient recognized the kindness, respect, and concern shown by the professionals during the appointment.

This caused patients to feel affection for the health personnel, and to be enthusiastic about returning for a consultation and/or examination because of all the collaboration, good treatment, and dedication during the appointment. It also reduced crowds, since, in a constant effort to protect the well-being of their patients, they did the examinations with enough time for them to leave and have a lower risk of contagion. Thus, recognition, good treatment, and the inclusion of patients in decision-making were reflected in perceived satisfaction.

## 4. Discussion

A group of RA patients who were evaluated using two models of health care (TM and UC) during the pandemic is analyzed in this study. After the first three months of confinement, some remained in the initial model, and others moved between models (mainly those who initially followed UC). An increase in self-care was found in the entire group. There was generally no change in disease activity variables no matter which model they used, although the predominant baseline level of activity was mild, moderate, or in remission in the whole group. The analytical methodology used that included quantitative and qualitative analysis is novel for RA patients under pandemic conditions.

Given the high risk of overloading intensive care units, strict confinement and mobility restrictions were adopted in different countries [33,34]. Patients with chronic conditions were advised to limit leaving their home, including clinic visits [35]. This led to physical outpatient consultations being replaced by teleconsultation to ensure adequate continuity of care [33]. George MD et al. [18] surveyed American patients with common rheumatological diseases electronically, and found that of 925 patients with RA (56.6%) avoided going to the doctor’s office, and 29.5% had a telemedicine visit. Ciurea A et al. [36] showed that in a Swiss cohort of patients with rheumatological conditions, the number of consultations dropped by 52%, whereas the number of remote assessments increased by 129%.

Some researchers, like us, made remote visits exclusively through telephone ICT. López Medina et al. [34] evaluated the rheumatology patients’ level of satisfaction with phone consultation (0–100 scale). They found an intermediate to high level of satisfaction with this type of consultation (64.7 ± 35.8). In addition, 52.7% of patients considered it useful. In some settings [37], when evaluating HCP, it was found that two-thirds or more of the rheumatologists chose telephone follow-up for stable RA and other stable clinical rheumatological conditions. Going in depth, other researchers [4] found that during the first peak of the pandemic, the patients’ level of satisfaction with teleconsultation (including telephone) was high in proportion to their educational level, and the doctor’s satisfaction was greater when the patient had ICT management skills.

All these results can be contrasted with those found in our study through the qualitative approach, in which resilience mechanisms, such as adaptation to the consultation model and self-care measures to minimize risks from the pandemic, emerged in the patient groups. Regarding qualitative studies related to RA and teleconsultation in the emergency context due to COVID-19, teleconsultation management, including telephone and WhatsApp [4,38], during this pandemic has been carried out synchronously [39]. Asynchronous consultation, in turn, was done at the moment medical records were stored or sent. Furthermore, a study in which rheumatologists were trained to manage this model at the beginning of the pandemic was reported by the Special Surgery Hospital (SSH) [39] in the USA. The technology team was in charge of collaborating with the patients to download the Zoom application, through which these doctor visits were carried out. The qualitative results of that study suggested that training professionals in assertive communication, problem solving, and empathy, as well as training patients in digital literacy, could improve care in this model.

Like us, the group from Chevayard M [40] in Milan, Italy evaluated the outcomes of patients with rheumatological diseases under different treatments, and followed-up by phone during the 3-month lockdown. However, they used an online platform to capture patient-reported outcomes (PROMs), and did retrospective research. They found no statistically significant differences in general health on the visual analog pain scale, or other specific outcome measurements for patients with RA and other rheumatological conditions during follow-up. They concluded, as we did, that there was no increase in disease activity when patients were followed using TM. This must be taken into account and evaluated over longer periods of time, and at different levels of disease activity given that, in our study, disease activity was low or moderate for most patients.

The adherence patterns during the evaluation of RA patients through telemedicine are very relevant. Different results have been shown throughout the world during the height of the pandemic. Adult patients with rheumatological conditions were surveyed in the United States (mainly RA) [18] to assess concerns and healthcare-related behavior, and 14.9% of them showed a decrease in the use of DMARD. In contrast, researchers that evaluated inflammatory rheumatic disease (majority RA) patients in Germany [10] for three months of follow-up had only 4% of the patients report discontinuing their medication on their own, or in consultation with their rheumatologists. Furthermore, North American patients surveyed between April and May 2020 [41] had a higher likelihood of stopping medication with a telemedicine visit when compared to patients with an office visit. These results contrast with ours given that there was no change in adherence nor stopped medications among the patients that used telehealth exclusively during the total follow-up period. Conversely, those who used the UC model exclusively, and those who transited between the models during the follow-up period had a significant adherence decrease measured through MGLS. Mancuso et al. [42], showed that some medication modifications were suboptimal for disease control in rheumatological patients, but were made to mitigate infection risk and minimize potential harm when cessation of in-person office visits made laboratory tests and physical examinations unobtainable. The group of patients in our study who went from face-to-face to telemedicine may have decreased their adherence and methotrexate use due to the conditions contemplated by Mancuso et al. Note that there was a significant increase in self-care among patients who had their first visit (TR model) or the entire follow-up in the face-to-face model, probably related to the fear of contracting the infection due to leaving their home. The ASAS-R scale measurement of self-care used in this research is novel, and this is one of the first times RA patients have been evaluated using telemedicine.

Some authors [43] have shown that the level of rheumatological patients’ knowledge of the health authority guidelines on using medication for rheumatological diseases, as well as a high socioeconomic status (SES) [18] resulted in high or better adherence to, and continuation of, medications. In this study, the majority of the patients had a low SES and predominantly low and middle educational level, but these variables produced no differences when comparing the face-to-face vs. telemedicine group, nor had any impact on the adherence to medication level. In a study of Latin American patients [44] with rheumatological conditions, predominantly RA, it was found that 66.9% of the total number of patients had problems obtaining at least one of their medications, which forced them to discontinue.

Telemedicine has been used in many countries for years to improve [15] access to specialized care in rural areas, but that has not been the case here. This health care model is not totally implemented, so the present results could be seen as an opportunity to implement better care models, including telemedicine. The pandemic also provided the opportunity to try mixed health care models, and integrate face-to-face doctor visits with telemedicine [15,33]. Therefore, patients with stable rheumatic conditions, such as the ones in this study, could be followed by telemedicine for a period similar to the present. Replacing telephone consultation with more interactive telemedicine, including video calls and platforms for RA patient-monitoring with PROMs, would probably optimize therapeutic adherence, and produce better outcomes [33,45,46]. However, this would need to be implemented in regions with broad internet coverage and connectivity, as well as a high percentage of smartphone owners [33]. The educational level and SES are also variables to consider when including patients in TM in low- and middle-income countries such as ours. Likewise, disease activity, digital resources, and remoteness related to our geographical area should be considered. Depending on those variables, establishing a tele-rheumatology triage to better identify those patients who need urgent or in-person attention is a possibility, as some authors [33] have postulated.

The present study has some limitations. Participants’ voluntary transfer from one model to another prevented the analysis of all patients based on the group they belonged to at the beginning of follow-up. The follow-up period may not have been sufficient to determine if there were changes in some variables. Due to its characteristics, the DAS 28 scale could only be evaluated by a face-to-face consultation. In this study, only telephone follow-up was used, whereas others also used video calls, including PROMs. It was not possible to differentiate the contagion levels between the UC versus TM model, since the majority of patients who developed COVID-19 were transiting between models.

## 5. Conclusions

The use of telemedicine for rheumatic patients during the pandemic showed that it was as effective as face-to-face medical consultation in controlling RA activity levels, and helped these patients avoid infection risk.

The advantage was that patients had the option of moving between models, since, in the future, it may be necessary to develop mixed care models for RA to facilitate patient access to specialists. This model enables patients to improve their self-care skills, as seen in the significant improvement on ASAS-R.

Qualitative analysis supports this mixed care model, and shows acceptance of telemedicine by healthcare providers and patients. It suggests that mixed care models that consider clinical factors and social determinants may define the relevance of TM, and enable each patient and HCP to choose the most appropriate care model.

## Figures and Tables

**Figure 1 healthcare-09-01744-f001:**
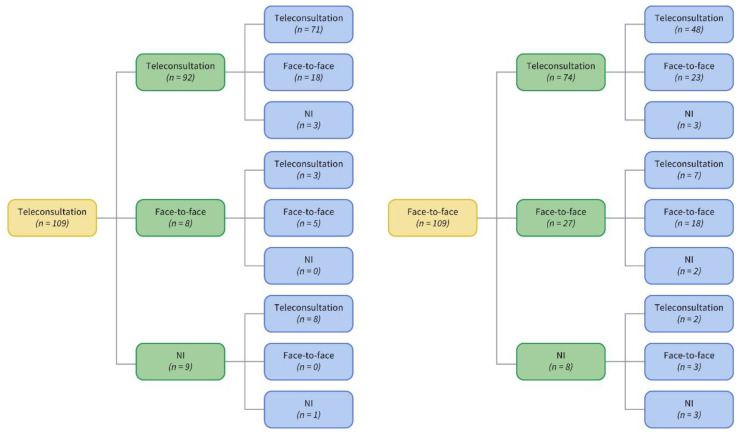
Patient flow chart at each follow-up (0–6–12 weeks). Yellow: baseline; green: 6 weeks; blue: 12 weeks. NI: no information.

**Table 1 healthcare-09-01744-t001:** Comparison of the general clinical data between Groups A and B at baseline.

Variable		Teleconsultation *n* = 109 Mean (Standard Deviation)		Face to Face *n* = 109 Mean (Standard Deviation)		*p* Value *
Age Age at onset Age at diagnosis		61.1	12.6	61.9	12.5	0.608
		47.7	13.6	46.7	13.5	0.593
		50.2	13.7	49.9	13.3	0.872
**Variable**		***n* (%)**		***n* (%)**		0.603
Sex	Female	90 (82.6)		87 (79.8)		
	Male	19 (17.4)		22 (20.2)		
Marital status	Married	44 (40.4)		51 (46.8)		0.055
	Single	34 (31.2)		21 (19.3)		
	Other	31 (28.4)		37 (33.9)		
Socioeconomic status (presential *n* = 106)	Low	61 (56)		58 (54.7)		0.794
	Middle or high	48 (44)		48 (45.6)		
Residence	Bogotá	77 (70.6)		89 (81.7)		0.57
	Outside Bogotá	32 (29.4)		20 (18.4)		
Occupational status	Household duties	46 (42.2)		48 (44)		0.000
	Intellectual/office work	18 (16.5)		4 (3.7)		
	Manual work	24 (22.0)		18 (16.5)		
	Other ^a^	21 (19.3)		39 (35.8)		
Educational level	Any	1 (0.9)		0 (0)		0.124
	Primary school	47 (43.1)		34 (31.2)		
	Secondary school	35 (32.1)		50 (45.9)		
	Technician	21 (19.3)		15 (13.8)		
	University	4 (3.7)		9 (8.3)		
	Postgraduate	1 (0.9)		1 (0.9)		
Comorbidities (Teleconsultation *n* = 108)	Arterial hypertension	36 (33.0)		38 (34.9)		0.775
	Osteoarthritis	82 (75.9)		86 (78.9)		0.600
	Fibromyalgia	2 (1.8)		11 (10.1)		0.010
	Hypothyroidism	27 (24.8)		25 (22.9)		0.751
	Osteoporosis	38 (34.9)		47 (43.5)		0.192
Previous surgical procedures		89 (84.0)		74 (67.9)		0.000
Erosivity		50/102 (49.0)		44 (40.4)		0.206
Extra-articular manifestations	Cutaneous	0 (0)		3 (2.8)		0.193
	Rheumatoid nodules	0 (0)		2/3 (66.7)		
	Digital ulcers and Raynaud’s phenomenon	0 (0)		1/3(33.3)		
	Pulmonary	4 (3.7)		1 (0.9)		0.600
	Pulmonary hypertension	3/4 (75)		0 (0)		
	Interstitial lung disease	0 (0)		1/1 (100)		
	Interstitial pneumonitis	1/4 (25)		0 (0)		
	Ophthalmological	0 (0)		1 (0.9)		0.499
	Episcleritis	0 (0)		1/1 (100)		
Polyautoimmunity	Sjögren’s syndrome	5 (4.6)		5 (4.6)		0.087
	Systemic lupus erythematosus	2 (1.8)		2 (1.8)		
	Systemic sclerosis	2 (1.8)		2 (1.8)		
	Other ^b^	9 (8.3)		2 (1.8)		
Previous Infectious history ^c^		8(7.33)		10 (9.17)		0.623
Medications at baseline						
Analgesics		80 (73.4)		77 (70.6)		0.651
Antimalarials		12 (11.0)		9 (8.3)		0.491
b/ts DMARDs		38 (34.9)		41 (37.6)		0.673
csDMARDS		99 (90.8)		97 (88.9)		0.653
GCs		79 (72.5)		72 (66.1)		0.304

Comparison of the general clinical and sociodemographic data between two models of attention. * Statistical differences were determined to be significant at *p*-values less than 0.05. ^a^ Other occupation includes: retired, self-employed, and unemployed. ^b^ Other polyautoimmunity includes: autoimmune thyroid disease, megaloblastic anemia, undifferentiated connective tissue disease, psoriasis, immune complex glomerulonephritis, and vitiligo. ^c^ Previous infectious included: epidural abscess, hepatitis C, herpes zoster infection, urinary tract infection, community-acquired pneumonia, osteomyelitis, and tuberculosis. b/ts DMARDs: biological/targeted synthetic disease-modifying antirheumatic drugs; csDMARDs: conventional synthetic disease-modifying antirheumatic drugs; GCs: glucocorticoids.

**Table 2 healthcare-09-01744-t002:** Changes in clinical outcomes and medications during follow-up of the whole group of patients.

	Visit 1 *n* = Frequency Median (IQR)	Visit 2 *n* = Frequency Median (IQR)	Visit 3 *n* = Frequency Median (IQR)	*p* Value *
VAS pain	*n* = 215 4 (2–6)	*n* = 201 4 (2–6)	*n* = 167 4 (2–6)	0.8382
PGA	3 (2–5)	3(2–5)	3 (2–4)	0.349
PAS	*n* = 215 3 (1.3–4.4)	*n* = 201 2.7 (1.4–4.1)	*n* = 168 2.7 (1.8–3.8)	0.8382
DAS28	*n* = 109 2.6 (2.1–3.6)	*n* = 35 2.7 (2.4–3.5)	*n* = 60 2.7 (2.2–3.5)	0.7115
HAQ	*n* = 216 0.07 (0–0.9)	*n* = 201 0.1 (0–0.6)	*n* = 175 0.1 (0–0.6)	0.1694
EQ5-VAS	*n* = 217 70 (50–80)		*n* = 206 70 (60–80)	0.1153
EQ5-overall index	0.7 (0.5–0.8)		0.7 (0.5–0.8)	0.4294
EQ5-TTO	0.7 (0.6–0.9)		0.7 (0.6–0.9)	0.411
ASAS-R	*n* = 218 65 (61–68)		*n* = 206 69 (64–77)	0.0001
MORISKY	Visit 1 *n* = 218 (%)		Visit 3 *n* = 206 (%)	*p* value *
Adherence	140 (64.2)		117 (56.8)	0.118
Non-adherence	78 (35.8)		89 (43.2)	
Medications				
	Visit 1 *n* = 218	Visit 2 *n* = 201	Visit 3 *n* = 206	*p* Value *
Analgesics	*n* (%)	*n* (%)	*n* (%)	
Acetaminophen	151 (69.3)	134 (66.7)	122 (59.2)	0.081
Codeine	7 (3.2)	9 (4.5)	12 (5.8)	0.429
Hydrocodone	30 (13.8)	26 (12.9)	23 (11.2)	0.715
Oxycodone	1 (0.5)	1 (0.5)	1 (0.5)	1.00
Tramadol	15 (6.9)	9 (4.5)	9 (4.4)	0.424
Antimalarials				
Chloroquine	19 (8.7)	15 (7.5)	12 (5.8)	0.938
Hydroxychloroquine	2 (0.9)	2 (1)	2 (1)	1.00
b/ts DMARDs				
Abatacep	2 (0.9)	3 (1.5)	3 (1.5)	0.824
Adalimumab	5 (2.3)	5 (2.5)	5 (2.4)	1.00
Certolizumab	22 (10.1)	17 (8.5)	18 (8.7)	0.827
Etanercep	15 (6.9)	14 (7)	13 (6.3)	0.959
Golimumab	12 (5.5)	11 (5.5)	12 (5.8)	0.985
Infliximab	5 (2.3)	5 (2.5)	4 (1.9)	0.945
Rituximab	3 (1.4)	2 (1)	4 (1.9)	0.776
Tocilizumab	9 (4.1)	9 (4.5)	9 (4.4)	0.984
Tofacinib	6 (2.8)	7 (3.5)	7 (3.4)	0.896
csDMARDS				
Azathioprine	4 (1.8)	4 (2)	3 (1.5)	0.915
Leflunomide	113 (51.8)	104 (51.7)	100 (48.5)	0.747
Methotrexate	127 (58.3)	116 (57.5)	96 (46.6)	0.027
Micofenolate	0 (0)	1 (0.5)	1 (0.5)	0.545
Sulfasalazine	36(16.5)	30 (14.9)	30 (14.6)	0.834
GCs				
Betamethasone	22 (10.1)	16 (8)	11 (5.3)	0.189
Deflazacort	10 (4.6)	11 (5.5)	12 (5.8)	0.841
Methylprednisolone	2 (0.9)	2 (1)	2 (1)	1.00
Prednisone	131 (60.1)	123 (61.2)	107 (51.9)	0.116

Changes in the whole group in clinical outcomes and medications during follow-up. * Statistical differences were determined to be significant at *p*-values less than 0.05. ASAS-R: The Appraisal of Self-Care Agency Scale-Revised; b/ts DMARDs: biological/targeted synthetic disease-modifying antirheumatic drugs; csDMARDs: conventional synthetic disease-modifying antirheumatic drugs; DAS28: 28-joint Disease Activity Score; EQ5-overall index: EuroQol5 overall index values; EQ5-TTO: EuroQol5 time trade-off; EQ5-VAS: EuroQol5 visual analogue scales; GCs: glucocorticoids; HAQ: health assessment questionnaire disability index; IQR: interquartile range; PAS: patient activity scale; PGA: patient global assessment; VAS: visual analogue scale.

**Table 3 healthcare-09-01744-t003:** Main clinical outcomes evaluated in three visits during follow-up.

Variable/Group	Median (IQR)	Median (IQR)	Median (IQR)	*p* Value *
VAS pain	Visit 1	Visit 2	Visit 3	
TM (visit 1 and 2 *n* = 71) visit 3 *n*: 62	5 (2–7)	4 (2–6)	3 (2–5)	0.1250
UC (visit 1 and 2 *n* = 18) Visit 3 *n*: 15	4.5 (2–6)	3 (2–5)	3 (2–6)	0.6935
TR (visit 1 *n* = 126 visit 2 (*n* = 112) visit 3 (*n* = 91)	4 (2–6)	4 (2–7)	5 (3–6)	0.5342
PGA	Visit 1	Visit 2	Visit 3	*p* Value *
TM (*n* = 71) Visit 3 *n*: 62	3 (2–5)	3 (1–5)	3 (1–4)	0.1203
UC (*n* = 18) Visit 3 *n*: 15	4 (2–5)	3 (2–4)	2 (2–4)	0.4411
TR (visit 1 *n* = 126) (visit 2 *n* = 112) (visit 3 *n* = 90)	3 (2–5)	3 (2–5)	3 (2–5)	0.9318
PAS	Visit 1	Visit 2	Visit 3	*p* Value *
TM (*n* = 71) Visit 1: 70 Visit 3 *n*: 62	2.8 (1.3–4.4)	2.7 (1.3–3.8)	2.2 (1.3–3.3)	0.2252
UC (*n* = 18) Visit 3 *n*: 14	3.2 (1.3–4)	2.7 (1.3–3.8)	2.5 (1.5–3.1)	0.7827
TR (visit 1 *n* = 126) (visit 2 *n* = 112) (visit 3 *n* = 90)	3.0 (1.6–4.4)	2.9 (1.5–4.4)	3.2 (2.0–4.1)	0.6796
DAS-28	Visit 1	Visit 2	Visit 3	*p* Value *
UC (*n* = 18) Visit 3 *n*: 14	3.3 (2.2–4.2)	2.4 (2.1–2.9)	2.6 (2.1–3.3)	0.1777
TR (visit 1 *n* = 91) (visit 2 *n* = 17) (visit 3 *n* = 45)	2.5 (2.1–3.6)	3.2 (2.5–3.6)	2.7 (2.2–3.9)	0.0542
HAQ	Visit 1	Visit 2	Visit 3	*p* Value *
TM (*n* = 71) Visit 1: 70 Visit 3 *n*: 61	0 (0–0.25)	0 (0–0.5)	0.13 (0–0.5)	0.3526
UC (*n* = 18) Visit 3 *n*: 15	0 (0–0.6)	0.2 (0–0.9)	0.13 (0–1)	0.7868
TR (visit 1 *n* = 128) (visit 2 *n* = 112) (visit 3 *n* = 99)	0.1 (0–1)	0.1 (0–0.69)	0.25 (0.1)	0.2504
EQ5 VAS	Visit 1		Visit 3	*p* Value *
TM (*n* = 71)	70 (50–80)		70 (60–80)	0.1199
UC (*n* = 18)	80 (60–80)		70 (60–80)	0.7954
TR (visit 1 *n* = 128) (visit 3 *n* = 117)	65 (50–80)		70 (50–80)	0.3385
EQ5-overall index	Visit 1		Visit 3	*p* Value *
TM (*n* = 71)	0.7 (0.6–0.8)		0.7 (0.5–0.8)	0.6077
UC (*n* = 18)	0.64 (0.6–0.8)		0.7 (0.5–0.8)	0.7747
TR (visit 1 *n* = 128) (visit 3 *n* = 117)	0.6 (0.5–0.8)		0.7 (0.5–0.8)	0.2503
EQ5-TTO	Visit 1		Visit 3	*p* Value *
TM (*n* = 71)	0.8 (0.6–0.9)		0.8 (0.6–0.9)	0.6049
UC (*n* = 18)	0.7 (0.6–0.9)		0.8 (0.6–0.9)	0.7747
TR (visit 1 *n* = 128) (visit 3 *n* = 117)	0.71 (0.5–09)		0.8 (0.6–0.9)	0.2686
ASAS-R	Visit 1		Visit 3	*p* Value *
TM (*n* = 71)	65 (60–69)		67 (63–71)	0.1481
UC (*n* = 18)	65 (62–67)		69 (67–78)	0.0077
TR (visit 1 *n* = 129) (visit 3 *n* = 117)	65 (61–67)		73 (66–79)	0.001
MORISKY	Visit 1 *n* (%)		Visit 3 *n* (%)	*p* Value *
TM (*n* = 71)				0.121
Adherence	39 (54.9)		48 (67.6)	
Non-adherence	32 (45.1)		23 (32.4)	
UC (*n* = 18)				0.006
Adherence	11 (61.1)		3 (16.7)	
Non-adherence	7 (38.9)		15 (83.3)	
TR (visit 1 *n* = 129) (visit 3 *n* = 117)				
Adherence	90 (69.8)		66 (56.4)	0.03
Non-adherence	39 (30.2)		51 (43.6)	

Main clinical outcomes evaluated during follow-up: clinical behavior and switching between models across the visits. * Statistical differences were determined to be significant at *p*-values less than 0.05; ASAS-R: The Appraisal of Self-Care Agency Scale-Revised; DAS28: 28-joint Disease Activity Score; EQ5-overall index: EuroQol5 overall index values; EQ5-TTO: EuroQol5 time trade-off; EQ5-VAS: EuroQol5 visual analogue scales; HAQ: health assessment questionnaire disability index; IQR: interquartile range; PAS: patient activity scale; PGA: patient global assessment; TM: telemedicine model; TR: transition model; UC: face-to-face usual care model; VAS: visual analogue scale.

**Table 4 healthcare-09-01744-t004:** Quotes from interviews of patients and health care professionals from four emerging categories.

	Fragments from Participants’ Comments
Factors present in communication	EUT25: “Well, as I said, communication is very important because we can tell the doctor what we have noticed.” EUT15: “Even though I tell him that it is inflamed, he [professional] cannot tell if it is or not. It is not the same as when he looks at it and realizes when he touches it that there is inflammation, and there is pain or that there may even be a fracture.” EP4: “Assertive communication. Since we are no longer using body language, we must start to improve communication to avoid what could sometimes be many mistakes.” EP1: “Empathy towards the patient, understanding the multiple difficulties he has when we are able to talk.” EP2: “Realize that there are other types of contrasts than black and white, and one has to grasp that parameter to see other ways. One wants to be objective and well, now everything is more subjective.”
ICT management	EUT22: “... Technology has already pushed us aside. It is very difficult to use the internet because I never learned how to. It is all I can do to answer this phone.” EP7: “All of this depends on the socioeconomic, sociocultural class of the patient if I have the opportunity to do teleconsultation” EP5: “If the patient has a hearing disability, does not have a companion, and the patient does not have access to technology.”
Family support and interaction	EUT23: “My daughter uploads them to the platform for me.” “My daughter is the one who does all this.” “here at home with my children and my great-granddaughters.” EP4: “Yes, we have found, I think, that the majority receive support from their extended family, and that makes things a little easier for a relative, friend, or a neighbor because all of this has to do with relatives, friends, neighbors who collaborate with them. Unfortunately, those who are alone do not have someone who can help them in this respect.” EP1: “we have this program, and it is to give patients an advocate.” “it is a program where, I think, there are 30 older adults, patients who do not have support networks”
Adherence to treatment	EUT3: “As commitments that someone tells you have to be taken on, you do so—such as medication, food, taking the medicine regularly, and I am happy in that sense. EP4: “Well, in terms of adherence, it has really improved. As I was saying, patients who have not had their check-ups for a long time can be reinstated. Then, if the adjerence of these patients increases a little, it could translate into a better prognosis. As for the impact, for example, reducing the number of visits to the emergency room or the number of hospitalizations, I couldn’t say if it is less in terms of face-to-face consultation, but yes, adherence improves.” EUT25: “For example, the doctor prescribed some medicine for me and… the truth is that I wasn’t able to get it for a long time due to difficulty with the platform, and until I had another appointment with him and he gave me the prescription again. I went without that medicine for more than a month. Only yesterday was I able to get it because the document also has to be authorized and that has caused many difficulties.”
Experiences of patients and healthcare personnel in COVID-19 time	EUT24: “Well, I do believe that the quality of life has changed for most of us because it has been more…. People have not been able to work or anything, and it has become more difficult anyway.” EUT18: “At times like this, at least, I am very afraid to go out.” EP1: “I know that COVID can influence anxiety, depression, stress, believing that the mass media is trying to make us doubt everything, being more careful to adapt ourselves because that is the most important word: adaptation.” EUT10: “Well, we have experienced it, ma’am, taking care of ourselves, going out wearing a mask, washing every time we come in from the street, putting alcohol on our shoes, leaving our shoes next to the door, washing our hands...”

Categories emerged by the subjects’ experience. EP: answer of health professionals; EUT: answer of patients treated in the telecare model; ICT: information and communication technologies.

## Data Availability

All data generated or analyzed during this study are included in this published article and in Appendix A.

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
