# Peer review of "Evaluation of a Non-Face-to-Face Multidisciplinary Health Care Model in a Population with Rheumatoid Arthritis Vulnerable to COVID-19 in a Health Emergency Situation†"

_healthcare, 2021, doi:10.3390/healthcare9121744_

Round 1

Reviewer 1 Report

Abstract:

The abstract is not clearly presented. It is not clear how the number of patients in group A decreased from 109 to 71. Also, it has not been mentioned that the group which transited between models has been divided into subgroups. The authors did not define the ">" sign.

"In Group A: (n=71), the increase in adherence was demonstrated."

Comment: This result was non-significant.

"In group B: (n=18), a significant increase on adherence level p= 0.06 and on the self-care capacity p= 0.0077 was found."

Comment: The adherence level decreased in this group.

Materials and methods:

Patients are classified as having rheumatoid arthritis if they fulfill the American College of Rheumatology classification criteria for rheumatoid arthritis. Please cite this reference.

Also references for PAS, DAS28,HAQ, EQ-5D-3L, MGLS, ASAS-R, Taylor-Bogdan proposal need to be cited.

How was the time trade -off calculated? Why isn't it mentioned in the results?

Sample size calculation:

It is mentioned that the sample size was increased to 110 while it is actually 109.

Results:

Table 1: The first raw in the table:

Median (standard deviation): It should be mean (standard deviation)

"Statistically significant differences were determined as p values <5%"

Comment: p values should be less than 0.05.

Figure 1:

What is meant by NI?

3.3.1 Telemedicine Model outcomes:

" 71 still remained in this modality by week 6."

Comment:  ...by week 12.

Table 3:

Patients were classified into 8 subgroups but only 5 subgroups with a sufficient number of patients were analysed. Analsis of many subgroups is confusing . It would have made more sense to classify the patients into three groups only:

Telemedecine group, face-to face group, transition model group. So, no patients would be left unclassified. 

Table 4:

There is no need for this table. It contains too many unnecessary details.

3.4. Qualitative results:

Unclear paragraph. The results need to be shown.

Discussion:

"There was generally a change in the disease activity variables.."

Comment: There was generally no change in disease activity variables...

"resilience mechanisms such as adaptation to the consultation model and self-care measures to minimize risks from pandemic emerged in the patient groups. Factors in communication, ICT management, and family support and interaction were relevant to both groups interviewed."

Comment: This should be shown in the results first using appropriate statistical tests.

"Telemedicine was also effective in maintaining low levels of contagion in RA patients, thus preventing greater morbidity and mortality. "

Comment: Please explain. Among the 7 patients who were infected, only one of the patients used usual care and the others used telemedicine.

"The qualitative analysis shows  acceptance of telemedicine by healthcare providers and patients."

Comment: This needs to shown in the results first using appropriate statistical tests.

Author Response

We thank the reviewer for the constructive criticism.

It greatly helped us to improve the quality of our paper.

We were able to use the suggestions and edited the manuscript accordingly.

Reviewer 2 Report

Overall, a well written paper evaluating a valuable tool for ongoing healthcare provision. There could be some improvement in the general use of language (e.g. line 47 - avoid saying "have been done"). 

The abstract is fair but there is a lack of reporting around the qualitative results. It would be beneficial to include the four categories that were derived. The last sentence is confusing - you have said that TM is favoured but that there were no major differences. I would consider rewording this to be more clear about whether one model was more favourable than the other. 

The introduction is a good summary of current literature and provides good context around the concepts and the local health area. Again, Line 77, avoid using the word "done". 

The methods are clearly described and there is evidence of ethical approval. It would be beneficial to include how participants were recruited. Line 143 - avoid saying "done". Do you have a reference for the Taylor-Bogdan proposal? Line 149. 

The quantitative results are presented very well. However, the qualitative results are lacking. Including a more detailed overview of your categories would strengthen the paper. For example, do you have any quotes to support the categories? Can you provide more detail about what each category means, why it was prominent in the discussions? It feels as though you do not value the qualitative inquiry. 

The discussion is good but again, there is very little evidence of including the qualitative findings. 

Your conclusions are nicely aligned with your aims, although including reference to your qualitative data would strengthen this as well. 

Author Response

We thank the reviewer for the constructive criticism.

It greatly helped us to improve the quality of our paper.

We were able to use the suggestions and edited the manuscript accordingly

Round 2

Reviewer 1 Report

The article has much improved , however, the part on qualitative analysis is not clear at all.

What was the aim of this questionnaire?

What were the questions included in this questionnaire?

What were the answers to these questions?

What was the level of satisfaction of patients to telemedicie?

What was the impact of the qualitative analysis on adherence to treatment and self care capacity of the patients?

All these details must be mentioned in a scientific article. The results must be presented using appropriate statistical methods.

The discussion is too long and the conclusion needs to be more concise.
